# A Systematic Review of Scientific Studies on the Effects of Music in People with Personality Disorders

**DOI:** 10.3390/ijerph192315434

**Published:** 2022-11-22

**Authors:** Rowan Haslam, Annie Heiderscheit, Hubertus Himmerich

**Affiliations:** 1Mental Health Studies Programme, Institute of Psychiatry, Psychology & Neuroscience, King’s College London, London SE5 8AF, UK; 2Department of Music Therapy, Augsburg University, Minneapolis, MN 55454, USA; 3Department of Psychological Medicine, Institute of Psychiatry, Psychology & Neuroscience, King’s College London, London SE5 8AF, UK

**Keywords:** music, music therapy, personality, personality disorders

## Abstract

Personality Disorders (PDs) are psychiatric conditions involving maladaptive personality traits and behaviours. Previous research has shown that musical preferences and the use of music may be related to personality traits. Additionally, music therapy is increasingly being used as a treatment option for people with PDs. Using the PRISMA guidelines, a systematic literature search was undertaken using three databases: PubMed, Web of Science, and PsycInfo. The following search terms were used: PubMed: “personality disorder” AND (music OR “music therapy”); Web of Science (advanced search): TS = (personality disorder) AND TS = (music or “music therapy”); PsycInfo: “personality disorder” AND (music OR “music therapy”). A total of 24 studies were included in this review and summarised into four categories: music preference, music therapy, music performance, and music imagery, all in relation to PDs or traits associated with PDs. The analysis found that individuals with personality traits associated with PDs may prefer different types or genres of music or interact with music differently than those without these traits. Additionally, music therapy (MT) was found to offer a potentially useful treatment option for PDs. The power of these findings was limited by the small number of included studies. This review offers a useful foundation upon which further research looking at MT as a potential treatment option for PDs can be built.

## 1. Introduction

### 1.1. Personality Disorders

Personality disorders (PDs) are conditions characterised by repeated patterns of maladaptive behaviours, thoughts, and inner states that negatively impact the individual’s quality of life [1]. These patterns cause impairments related to emotional regulation, impulse control, cognition, and relationships, which must be clinically significant to allow diagnosis [1]. PDs are associated with several poor outcomes, including high levels of mental health comorbidities, such as depression, anxiety, and substance misuse [2]; increased rates of mortality, both due to natural and unnatural causes [3]; and interpersonal difficulties [4].

The primary treatment options for PDs are psychotherapeutic and pharmacological interventions, although the latter is only recommended for treating specific symptoms [2]. The National Institute for Health and Care Excellence (NICE) [5] provides treatment guidelines for borderline personality disorder (BPD) and antisocial personality disorder (APD). The psychotherapeutic treatment options for BPD include dialectic behaviour therapy (DBT), cognitive behavioural therapy, schema therapy, and transference-focused therapy [6]. The NICE guidelines for APD recommend group cognitive behavioural sessions [7].

The large number of PDs and the vast scope of their corresponding symptoms makes research and treatment challenging. There are 11 PDs described in the Diagnostic and Statistical Manual of Mental Disorders [1] (including personality disorders not otherwise specified) organised into three clusters: Cluster A—odd/eccentric; Cluster B—dramatic/unpredictable; Cluster C—anxious. The ICD-11 has discarded PD categories and instead presents a dimensional model whereby individuals are diagnosed as having a “mild”, “moderate”, or “severe” PD [8]. In both the categorical and dimensional systems, PDs are heterogeneous, and the unique ways they present in individuals pose a barrier to effective research and developing national treatment guidelines [9]. Additionally, the different types of PDs and their high levels of comorbidities make it logistically challenging to isolate specific traits to research effective treatments [10]. Research on PDs has mainly focused on BPD and APD, limiting our understanding of effective treatments for other PDs [2]. Although some treatment options for PDs have shown promising results [11,12], such limitations mean there is still much to learn about effectively treating PDs. Given the severity and relative ubiquity of PDs, which impact individuals’ quality of life and put significant pressure on mental health services [13], further research into effective treatments of PDs is essential.

Personality traits are related to an individual’s characteristic way of thinking and behaving. Differences between individuals’ personalities are present from birth and arise from a combination of genetic and environmental factors [14]. PDs can be considered as existing on the extreme end of the spectrum of normal personality traits, and a PD diagnosis may be appropriate when the individual’s personality traits cause distress for the individual and those around them [14]. Specific personality traits which are associated with different PDs include suspiciousness, impulsivity, insecurity, deceitfulness, narcissism, and emotional dysregulation [1]. As these traits and others associated with PDs are commonly found in the general population, the point at which a diagnosis of a PD should be made is a challenge, and it is hypothesised that many more people have PDs than those that currently have a diagnosis [8].

An additional difficulty in diagnosing PDs is the point at which a diagnosis should be made in an individual’s lifespan. Personality traits, including those related to PDs, are more flexible across lifespans than previously thought [14], and maladaptive traits may be improved with appropriate intervention [2]. A PD diagnosis, however, is not usually given to those under the age of 18—in the case of APD, a diagnosis before 18 is not allowed—although this is a controversial decision. The ICD-11 allows earlier diagnosis where appropriate, with the objective of treating individuals as early as possible [8]. With these considerations in mind, the authors of this review chose to include studies that examined traits related to PDs and those that included participants under the age of 18 with the intention of making this review as broad as possible. As the definition, diagnosis, and categorisation of PDs have changed over time, we included all articles in this review where PDs were defined according to the DSM-III-R, DSM-IV or DSM-5, ICD-10, or ICD-11, and all articles where traits of PDs were defined according to one of the above-mentioned diagnostic manuals. 

### 1.2. Music Preferences, and Music Perception in PDs

Previous research has found associations between personality traits and a preference for different types of music [15]. As personality traits can be considered part of a spectrum, with maladaptive personality traits representing an extremity of normal traits, it is logical that PDs or personality traits related to PDs may also be associated with a preference for certain types of music. Research has found, for example, an association between aggressive music genres and reduced empathy in men [16]. Additionally, several researchers have examined the ways in which people with PDs or maladaptive personality traits perceive and respond to music. Strehlow and Lindner [17] found that some individuals with BPD utilised music as an emotionally protective tool, whereas others avoided music altogether due to the emotional vulnerability felt when listening to music. Kenner and colleagues [18] found that participants with BPD were emotionally overwhelmed by music and could identify this state as being connected to their BPD.

### 1.3. Music Therapy and PDs

Music therapy (MT) is the use of music by trained and credentialed professionals to provide benefit to an individual through enabling treatment, enhancing their recovery, or generally improving their well-being [19]. Despite being traditionally regarded as an alternative or complementary therapy, research findings that indicate the benefits of MT for different psychiatric conditions are increasingly common [20], including for psychotic illnesses [21]; autism spectrum conditions [22]; and obsessive-compulsive disorder [23]. Music is unique in its ability to generate and intensify different types of emotions [24], and it therefore offers an exciting area of potential therapeutic benefits for people with emotional difficulties. The National Health Service [25] offers MT (and arts therapies in general) as a treatment option for BPD. There is, however, a limited amount of quantitative data in this area [26], in part because only relatively recently have researchers considered MT a useful treatment option for people with PDs [27]. Initial research has shown some positive results, with MT increasing an individual’s resilience by increasing positive emotions [28] and providing individuals with means to express their emotions in a healthy way [29].

### 1.4. Aims and Objectives

To the authors’ knowledge, no previous systematic review has been undertaken on this topic. This systematic review aims to summarise the findings of the existing literature on music, music therapy, and PDs, and to identify any gaps in the existing literature.

## 2. Materials and Methods

### 2.1. Search Strategy

This systematic review followed the Preferred Reporting Items for Systematic Reviews and Meta-Analyses guidelines [30]. Following initial scoping searches, PubMed, Web of Science, and PsycInfo (Ovid) were chosen to extract the studies for this review. Given the limited literature published on music and PDs, the search terms were deliberately vague to capture as many studies as possible [31]. The Boolean operator “AND” was used to combine the search terms and increase the sensitivity of the search. The following search terms were used: PubMed: “personality disorder” AND (music OR “music therapy”); Web of Science (advanced search): TS = (personality disorder) AND TS = (music or “music therapy”); PsycInfo (Ovid): “personality disorder” AND (music OR “music therapy”). The first author conducted a supplementary backward reference search but, due to the limited literature in this area, this yielded no additional studies. No other handsearching was undertaken.

### 2.2. Inclusion and Exclusion Criteria

The eligibility criteria were designed to capture studies relevant to the aims of this systematic review [31]. These were as follows:

Inclusion Criteria:Studies were published in English or had an English-language abstract.Studies were original.Study participants were diagnosed with, or had symptoms related to, a PD.Study participants were asked about music; music or MT was used as an intervention; or participants experienced music-related symptoms.

Exclusion Criteria:Animal studies.Studies using only unoriginal data.Case studies, conference articles, systematic reviews, or meta-analyses.Studies that did not report results or clinical outcomes.Studies on dance or art therapy that did not report a specific result for MT.Studies where music was not linked to PDs or related personality traits.Studies where the results for participants with PDs or traits related to PDs were not reported separately.Studies that examined only a particular piece of music, composer, or performer.

### 2.3. Screening, Extraction and Quality Assessment

Studies found during the searches were imported into EndNote. Duplicate studies were removed firstly by the software and then the first author manually removed remaining duplicate studies.

A two-stage screening process was used, comprising an initial title and abstract screening and then an in-depth screening. In the first stage, the first author extracted all the abstracts of the potential studies from EndNote to an Excel spreadsheet. Two of the authors (RH, HH) independently reviewed the titles and abstracts to check whether they met the eligibility criteria. In the second stage, the first author manually extracted relevant information (citation details; sample characteristics; study design; assessment tools and questionnaires; types of treatment; main outcomes; and statistical significance) from the studies that met the inclusion criteria after the first stage. Two of the authors (RH, HH) then independently used the extracted information (and full text when necessary) to check the validity of each study against the JBI Global Critical Appraisal Tools [32]. All studies that met at least six of the quality assessment criteria were included in this review. The authors contacted the author of one study [33] to access the full article, as only the abstract was available online. The full article was only available in German and was translated by the first author (RH) using Google Translate. The other reviewed studies were all in English.

### 2.4. Analysis

For the narrative synthesis to summarise and compare the included studies, Popay and colleagues’ [34] guidelines for narrative synthesis were used. The included studies were summarised in tables and organised according to theme, which allowed for collation of articles with similar research questions or aims; they were later compared and discussed within their specific area of research.

## 3. Results

Using the above search methods, a total of 185 papers were identified, which was reduced to 171 papers after duplicate articles were removed. Figure 1 depicts the PRISMA Flow Diagram of the selection process [30]. Table 1 depicts a summary of all included studies (*n* = 24).

### 3.1. Studies Looking at Music Preference in People with PDs or Personality Traits Associated with PDs

#### 3.1.1. Studies Looking at Music Preferences in People with PDs

Gebhardt and colleagues [35] found that participants with personality and behavioural disorders (category F6 of the ICD-10, German-language version) used music mainly for cognitive problem solving and for reducing negative emotional arousal. Compared to participants with other psychiatric diagnoses, those with personality and behavioural disorders and schizophrenia most often used music as a tool for relaxation. All psychiatric participants used music for emotional regulation more than the healthy controls.

#### 3.1.2. Studies Looking at Music Preferences in People with Personality Traits Associated with PDs

Bowes and colleagues [37] investigated the associations between personality traits and music and film preferences. Of the results relevant to this review, they found a moderate association between openness to experience and a preference for jazz and blues, and a weak association between openness to experience and a preference for rock and alternative music. Correlations between personality factors and entertainment preferences were generally stronger in relation to film preference than for music preference.

Another study that examined personality and music preferences included traits related to autism in their analyses [42]. Young participants (aged 35 and under) completed questionnaires related to music behaviours, personality traits, and traits related to autism. The analysis found that people with higher levels of extraversion and fewer autistic traits reported greater levels of emotional responsiveness to music.

Three studies examined responses to different music genres. Gerra and colleagues [39] measured neuroendocrine changes and changes in the emotional state of participants after listening to classical or techno music. Following the experimental trial, it was found that while techno music increased stressfulness in participants, those with novelty-seeking temperaments experienced a reduced negative effect of listening to techno music compared to those without novelty-seeking temperaments.

Schwartz and Fouts [41] were concerned with music preferences, personality in adolescents, and developmental problems. Participants completed questionnaires related to personality, developmental problems, and music preferences. The authors found that participants with a preference for heavy music qualities (such as “wild and violent” and “loud, played at a great volume” [41] (p. 208)) were more assertive, moody, disrespectful to others, impulsive, and had lower self-esteem compared to those with a preference for light music qualities (such as “mild and quiet” and “soft and tender” [41] (p. 208)) and eclectic music qualities.

Merz and colleagues [24] recruited 400 participants to complete a series of questionnaires. They found that a preference for intense or aggressive music (e.g., rock, punk, and heavy metal) was non-significantly correlated with higher scores on all the aggression questionnaires. As this result was not significant, it could not be concluded that a preference for intense music was correlated with higher aggressive behaviours.

Another study explored sad music [40]. Participants were asked to complete personality questionnaires and a questionnaire related to their experience of listening to sad music. The analysis showed that conscientiousness, emotional stability, and empathic concern were all significantly negatively correlated with the employment of sad music in positive situations involving others, such as for a celebratory event, whereas “openness to experience” was significantly positively associated with the use of sad music. Rumination was associated with enjoying sad music but not with using sad music in positive situations involving others.

Gallagher and colleagues [38] were interested in the use of music for people with obsessive-compulsive personality traits. Participants were divided into three groups: an obsessive-compulsive personality group, an avoidant personality group, and a control group. Before completing cognitive ability tests, they were allowed to listen to music or listen to information about the test (thus reducing uncertainty). The obsessive-compulsive group spent less time listening to music and more time listening to the information tape when compared to the avoidant personality and control groups.

Finally, Garralda and colleagues [36] measured the heart rate and skin conductance of children during mental tasks and grouped them by diagnoses of emotional and conduct disorders (ICD-9) and by symptoms of neurosis and antisocialism. There was no significant difference in the heart rate or skin conductance between the symptom or diagnosis groups.

### 3.2. Studies Looking at MT for People with PDs or Personality Traits Associated with PDs

#### 3.2.1. Studies Looking at MT for People with PDs

Five studies investigated MT for participants with BPD. Foubert and colleagues [44] used a semi-structured instrumental (piano) improvisation method to investigate interpersonal functioning in participants with BPD. All MT sessions were recorded and, following the completion of all the sessions, the results were analysed musically and statistically, with the statistical analysis relating to the rhythmic timing of the participants’ improvisation, which the authors suggested related to interpersonal effectiveness. The statistical analysis showed that temporal lags in participants with BPD, whereby participants struggled to match the metronomic timing of the accompanist, predicted a diagnosis of BPD with 82% success.

Another study led by Foubert [45] made use of the same semi-structured instrumental (piano) improvisation method with participants who had been diagnosed with BPD prior to the study. Over the course of the sessions, six patterns of inhibited musicality were identified: reduced musical flexibility and spontaneity; diminished musical creativity; limited exploration and expansion of musical motifs; disrupted musical synchronicity; relying heavily on and copying the music therapist; and disconnected or random melodic and rhythmic generation.

Kenner and colleagues [18] conducted a study in the outpatient clinic of a private psychiatric hospital with patients who either had a diagnosis or traits of BPD and who had previously completed DBT therapy at the hospital and had since been referred for group MT. The MT sessions were filmed and analysed for various factors, including speech, musical improvisations, body language, and facial expressions. Over the course of MT, participants grew more confident in their handling of instruments and improvising, and their improvisations became more rhythmically synchronous.

Plitt [33] conducted a qualitative study examining implicit relations, defined as patterns of mood, cognition, and gestures that are learnt from the mother in early childhood, in MT for participants with BPD. Plitt’s analysis showed that the improvisation/conversation framework could be helpful in developing communication between participants and therapists and that the shared improvisation process helped to foster a relationship between both parties.

Strehlow and Linder [17] examined the patterns of behaviour that are typically present in MT sessions for patients with BPD. The authors used a systematic qualitative method to analyse the significant interactions within the MT programme for each participant. The authors’ analysis revealed ten key interactions for patients with BPD in MT, including the patient’s refusal to play; the patient withdrawing into the music; and the patient seeking structure through the musical interaction. Understanding these typical interactions may be beneficial for individuals with BPD seeking to engage in MT.

Chwalek and McKinney’s study [43] comprised two phases. The first phase involved a quantitative survey related to music therapists’ use of DBT in mental healthcare settings. Due to limited responses, the second phase comprised a semi-structured interview with two music therapists regarding their use of DBT. The results of the survey found that 38.3% of participants used elements of DBT in their music therapy practice. Of these, the majority used DBT for clients with BPD to address mindfulness, emotional regulation, and tolerating distress, rather than interpersonal effectiveness.

Gebhardt and colleagues [26] recruited participants with various psychiatric diagnoses (PDs = 3% of total) to compare the use of music and personality factors between two groups: those who received MT and those who did not. Participants in both the MT and non-MT groups completed questionnaires related to personality and musical dimensions, and subsequent analysis showed that in the MT group, personal insecurity predicted the use of music for problem solving and for seeking fun. Only two participants with PDs were included in both the MT and the non-MT groups.

Pool and Odell-Miller [29] conducted a qualitative mixed methods study, comprising a single case study of a patient with antisocial and avoidant PDs (and traits of BPD) and a qualitative interview with three music therapists. The results of the interview analysis revealed several points: aggression is a component of creativity; aggression needs a trigger, which can include group dynamics (which is significant in the context of group MT); and MT can provide a safe and productive way for an individual to express aggression.

Finally, in a one-year, naturalistic follow-up study, Hannibal and colleagues [46] examined whether participants with a PD or schizophrenia diagnosis adhered to MT treatment. They collected demographic and clinical data of psychiatric patients in Denmark who had been referred to MT programmes by their psychiatrists. The results found that adherence to MT for participants with PDs was high (87%) and that none of the participants’ demographic factors (e.g., age, gender, education) predicted adherence.

#### 3.2.2. Studies Looking at MT for People with Traits Associated with PDs

Montello and Coons [48] sought to determine whether active MT (making music) and passive MT (listening to and discussing music) showed different effects in preadolescents with emotional, behavioural, and/or learning disorders. Participants were assigned to groups receiving either passive or active MT over two 12-week periods. Group B’s hostility scores were significantly reduced following active MT while Group A’s hostility was significantly increased after active MT and non-significantly reduced after passive MT. Group A’s hostility scores were lower than Group B’s before beginning MT, suggesting that the groups may not have been similar to begin with. Qualitative data obtained after the MT intervention found that Group A contained the most insecure members of all the groups and that members required some time to get used to MT before they could benefit from it and, subsequently, improve their hostility scores.

Hunter and Love [47] conducted a case series examining the effects of health and safety interventions at a maximum-security forensic psychiatric hospital. Employing the Total Quality Management FADE (focus, analyse, develop, executive) technique, a multidisciplinary team made several recommendations to reduce mealtime violence, including playing music during mealtimes selected by the resident music therapists. A year following the changes, it was found that violent incidents in the dining rooms were reduced by 40%—however, this cannot be attributed solely to the introduction of music at mealtimes, as this change was implemented alongside several other changes to improve mealtime safety.

Ziv and colleagues [49] investigated the effects of music relaxation compared to progressive muscle relaxation (PMR) in older adults with insomnia. Comparing the effects of music relaxation and PMR, the analysis revealed that the lower a person’s agreeableness score, the greater improvement they reported in the number of hours they slept when using music relaxation, and the higher their extraversion score, the greater their improved sleep efficiency with music relaxation (although extraverts also benefited from PMR). Overall, the music relaxation method was found to be more effective in improving sleep than the PMR method.

### 3.3. Studies Looking at Music Performance in People with Personality Traits Associated with PDs

The one study that explored music performance in relation to personality traits associated with PDs [50] gave questionnaires related to personality and music experiences to performance students in the US. The subsequent analysis revealed that students who trained in non-classical music had significantly higher empathy scores than those studying classical music and those who studied both genres. Additionally, those who regularly played in small group ensembles were found to have significantly higher empathy scores. The authors hypothesised that non-classical music could involve a higher level of collaboration than classical music and that small ensembles promote skills related to empathy and interpersonal effectiveness.

### 3.4. Studies Looking at Music Imagery in People with Personality Traits Associated with PDs

Finally, two studies investigated involuntary music imagery (INMI) and personality traits associated with PDs. Mullensiefen and colleagues [51] recruited participants to complete questionnaires related to obsessive-compulsive traits and to INMI, otherwise known as “earworms” (the experience of hearing music without the presence of external stimuli). Analysis revealed that high levels of most obsessive-compulsive traits were positively related to the frequency of, and disturbance caused by, INMI; however, they were only indirectly related to the length of involuntary music and imagery episodes and the negative effect of episodes. Additionally, subsequent analysis revealed that the participants represented a subclinical population.

The last study by Negishi and Sekiguchi [52] also used a cross-sectional design to examine INMI in relation to personality traits. Japanese university students were given personality measures and taught how to use the experience sampling method for measuring their episodes of INMI; they were given frequent reminders on their smartphones to report whether they had experienced INMI and to record the emotional characteristics of the episode. After a seven-day period, the data were collected and analysed, revealing a significant positive association between intrusive thoughts and the occurrence of INMI. A summary of the main relevant results from this review can be found in Table 2.

## 4. Discussion

### 4.1. Methods and Content of Included Studies

This systematic review aimed to investigate the possible links between music and PDs. This aim was deliberately broad and allowed for a wide scope of literature to be reviewed, including literature that examined traits related to PDs and those that included participants under the age of 18. The literature search generated 24 studies with a total of around 5470 participants included in the final analysis (the exact figure is uncertain as the study by Hunter and Love [47] included a prison population with a maximum of 1000 participants). The studies retrieved by the search were summarised into four broad categories: music preferences or responsiveness; MT; music performance; and music imagery, all in relation to PDs and personality traits associated with PDs.

Of the nine studies that examined PDs or personality traits associated with PDs, the heterogeneity of the studies means that no overarching summary related to specific traits or musical features could reasonably be made. The results of individual studies, however, provide interesting evidence to suggest that some specific personality traits, such as reduced self-esteem [41]; conscientiousness [40]; openness to experience [37,40]; extraversion [42]; obsessive-compulsiveness [38]; and novelty seeking [39], may impact an individual’s response to, use of, or relationship with, music. Not all included studies, however, found significant associations between personality traits and a particular musical preference [24,36], suggesting that some personality traits and musical genres may be more interrelated than others. This is consistent with a recent meta-analysis looking at personality traits and musical preferences, which did not find any significant relationships between the Big Five personality traits and musical preferences [53]. While a significant association was found between assertiveness and a preference for heavy music qualities in adolescents [41], a nonsignificant association between aggression and heavy music genres was found in adults [24]. Additionally, one study found that participants with PDs use music primarily to aid cognition and reduce negative emotions [35].

Four of the included studies found that the therapeutic use of music could be beneficial for people with PDs or maladaptive personality traits [18,29,47,48]. Six studies found that MT poses unique factors and challenges in patients with PDs compared to other conditions [17,18,26,33,44,45]. In one study, while MT was generally effective in reducing children’s hostility scores [48], active MT initially increased some children’s hostility scores, suggesting that MT is not always a useful or benign intervention but can, in some cases, intensify maladaptive qualities. One study [43] found that DBT skills are not typically embedded in MT practice for psychiatric populations. Another study [46] found that participants with PDs mostly adhered to MT sessions. Finally, one study [49] found that music relaxation could be beneficial to those with lower levels of agreeableness and those with higher levels of extraversion, supporting the idea that different personality types respond differently to music. These results suggest that MT could be a useful treatment option for people with PDs or maladaptive personality traits. MT, however, should be tailored to address the different therapeutic needs of individuals, as the same MT interventions do not work equally well for all people and in some cases may be unhelpful.

Three studies did not fit with our initial aims of looking at music preference or MT in those with PDs. Two of these studies investigated INMI and found that intrusive thoughts were associated with the occurrence of INMI [52] and that high levels of obsessive compulsiveness were associated with INMI frequency and disturbance [51]. The last study looking at music performance found that non-classically trained music students and those who performed regularly in small ensembles were significantly more empathetic than classical music students and those who did not perform in small groups [50]. This may be due to the high levels of interpersonal effectiveness required to play with others in small ensembles (e.g., watching others’ movements and engaging in synchronous rhythms and dynamics) and that small ensembles are frequently used in pop and jazz music. Group MT interventions may therefore be beneficial for promoting interpersonal effectiveness compared to 1:1 MT. This idea was explored by Kenner and colleagues [18] who found that participants’ improved musical competency in group MT encouraged them to gain insight into working with others. Given that relational difficulties are a significant component of PDs and that these difficulties are not usually addressed in MT [43], this is a research area that warrants further development.

One finding from this review is that individuals may respond differently to different genres of music and, therefore, different methods of MT. Given this finding, it is notable that none of the included studies sought to determine whether personality traits, other psychological features, or biomarkers can predict an individual’s response to MT. As specific symptoms, such as intensive depressive symptoms, have been found to influence the treatment outcomes for other PD treatments, for example, schema cognitive behavioural therapy [54], it is plausible that such symptoms could impact the effectiveness of MT for PDs.

### 4.2. Future Directions

Further questions raised from this review concern whether certain personality traits increase an individual’s preference for certain genres of music, whether engaging with certain types of music amplifies or encourages certain personality traits, or whether there is a bidirectional relationship between music and personality. It is likely that different music preferences and different personality traits affect individuals in different ways due to several complex factors which cannot be explained by a single personality trait [24]. A preference for a certain type of music may be also due to upbringing, culture, education, friends and family preferences, accessibility, performance opportunities, and other interacting factors. It is also important to note that these factors which shape an individual’s understanding of music, music genres, and their preferences are not consistent among all fans of a certain genre. For example, there is no single age profile that enjoys jazz music, suggesting that not all fans of jazz enjoy it for the same reasons [55]. As people are accessing the same music for different reasons, one could not reasonably conclude that a single personality trait encourages all fans that prefer the same genre of music. Additionally, it is important to note that one’s music preferences fluctuate as their cultural status changes [55], meaning that an individual’s reason for liking a particular genre of music may change over time. These factors are important to consider as they may limit the clinical implications of studies that look at personality traits and music preferences without considering the myriad of variables that contribute to developing one’s music preferences. Additionally, they may also impact how an individual can engage in MT given their preferences, cultural context, and musical knowledge.

Future research could seek to investigate some of these questions by utilising cohort studies to document personality preferences and the perception and use of music across childhood development. Such studies may explore the ways in which music is processed in the brain, specifically the features of music to which we attach meaning, memory, and imagery. These features, which are shaped by several complex factors, influence our anticipation and subsequent enjoyment of music [56].

There are several recommendations for clinicians and researchers to advance the research related to the use of music and MT for individuals with PDs. Given the unique nature of music preference, it may be helpful to conduct clinical case study research examining the preferred music of individuals with PDs and the functions music serves in their day-to-day lives [57,58,59]. Comparing case studies or a series of case studies can help inform best practices and foster future research recommendations. Furthermore, it may be helpful to conduct phenomenological studies to understand the experiences of individuals with PDs in MT, or with different types of music interventions. This can help clinicians and researchers understand if music is being used in healthy and/or unhealthy ways [60]. Further understanding a client’s experiences and use of music and MT can help to inform and support in determining aims for outcome-based research.

### 4.3. Limitations

Reporting music interventions can be challenging due to the specific nature of music and the lack of adherence to guidelines [61]. Several studies did not provide consistent MT sessions or did not report on the specific type of MT methods being used [18,26,33,46,48]. Two studies either did not specify the type of music being used or justify why a particular genre/piece of music had been chosen [47,49]. Failing to report on the details related to music interventions limits the replicability of studies and the ability to extract, build upon, and apply relevant findings to clinical practice.

While every effort was made to preserve academic integrity in this review, the intrinsic subjectivity of a narrative review means that an objective standard is never completely met. By providing a clear methodology and organising the results in a logical and ordered way, in accordance with guidelines, the authors have sought to provide as clear and replicable a review as possible. It is the case, however, that any replication of this study may result in different interpretations of the studies and their results, and acknowledging this fallibility is important for those seeking to build upon the results of this study.

An important point to consider is that confounding variables may have influenced the results of the included studies. This review did not systematically check for confounding variables due to the spread of study designs, research topics, and participant groups. Of the studies that found results, several identified age and gender as potential confounders and subsequently accounted for this in their analyses (e.g., [26,35,37,41]). Another study found associations with ethnicity and empathy and controlled for this in their analysis [50]. Some studies did not explicitly address potential confounding variables, such as age or gender, in their analyses (e.g., [39,51]). Previous research has found associations between music preference and several variables, including age, ethnicity, gender, and social and occupational class [55]. We would therefore recommend future studies looking at PDs and music to consider these variables in their analyses to ensure that any conclusions that are drawn are not misrepresenting the effects of the variables. In the context of this review, the summarised results should be understood in the context of potential confounders.

The studies on music preference and the use of music were cross-sectional. While this is useful to reveal potential correlations between certain factors, causation cannot reasonably be drawn from the findings of these studies. For example, it is unclear whether young people with aggressive personality traits become aggressive because of listening to heavy music [41] or whether heavy music qualities are preferred by aggressive adolescents as it reflects their internal state.

Additionally, several studies used qualitative, descriptive, or quasi-experimental methods which, while useful, do not provide robust evidence for clinical application. Furthermore, several studies were limited by a small sample size (*n* < 30: [17,18,29,33,36,39,44,45,48,49]). While this may be reflective of the nascency of the field, the lack of randomised controlled trials limits the clinical application of the findings from this review.

A limitation in this research area more broadly is the logistical difficulty of studying PDs and personality traits related to PDs. The spectrum of PDs covers a wide range of maladaptive personality traits and behaviours, many of which are present at a subclinical level in the wider population. Isolating specific personality traits in relation to PDs is complex, in part due to the vastness of traits that can be considered related to PDs and the challenges with diagnosing PDs more broadly. This review included studies that examined traits related to PDs as well as studies that included participants under the age of 18 with the intention of making it as broad a review as possible. Looking at subclinical traits may be beneficial for understanding the spectrum of maladaptive personality traits, and may therefore inform our understanding of PDs, but extrapolating results from subclinical traits and applying them to clinical populations should be approached with caution.

Finally, this specific area of research is new, and this systematic review was deliberately broad in scope to incorporate all the relevant literature. Due to this and the small number of studies included in this review, the resulting studies were highly heterogeneous and there are significant gaps in the literature. Additionally, because the purpose of this study was to provide an initial overview of the literature on PDs and music, we were not able to formulate a specific research question. Ideally, a future systematic review following a similar methodology would benefit from a greater wealth of relevant literature.

## 5. Conclusions

This review identified four links between music and PDs: music preference, MT, performance, and music imagery. These findings provide a useful foundation to build upon for future researchers seeking to expand these research areas. In relation to MT, this study has provided some evidence to suggest that MT may be a useful treatment option for PDs. While this is something that is available in the UK for individuals with BPD, people with other PDs are not routinely offered MT as a treatment, partly due to the lack of treatment guidelines for PDs other than BPD and APD. Building upon previous research looking at personality, the studies looking at personality traits have demonstrated that people with PDs or specific personality traits may interact with music differently from others, but the strength of these relationships is underdetermined. As demonstrated in this review, MT has the capacity to be unhelpful for certain individuals, warranting further research into understanding the optimal uses of MT for different populations. While this is a relatively new area of research, the potential benefits of MT shown at this stage offer an exciting opportunity for future research and treatment for people with PDs and those with maladaptive personality traits.

From a clinical perspective, this review article identified several ways to use music in order to support people with PDs (see Table 2): as selected music has been reported to help to reduce violence and hostility, patients may develop playlists with the support of their therapists to manage aggression and violent impulses in foreseeably difficult situations; appropriate music for bedtime relaxation can be recommended to improve sleep length and quality; and people who experience insecurities may be encouraged to try music to aid cognitive problem solving and improve their mood.

## Figures and Tables

**Figure 1 ijerph-19-15434-f001:**
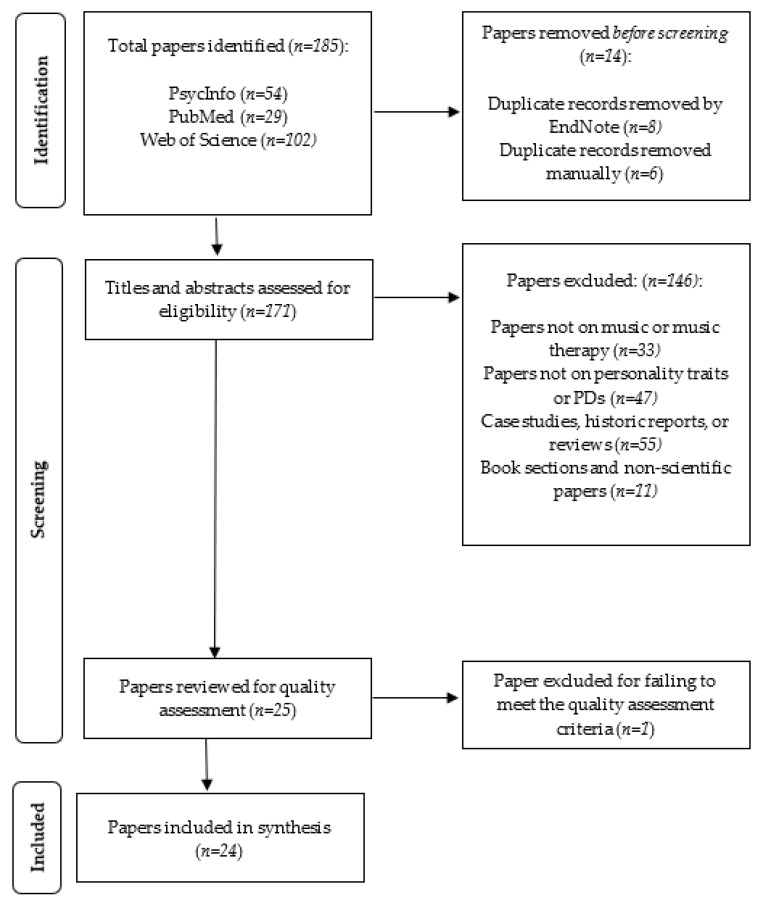
Prisma Flow Diagram.

**Table 1 ijerph-19-15434-t001:** Summary of included studies (*n* = 24).

Authors (Year)	Country	Sample and Group Size (*n)*	Total *N*	Age Range	Mean Age	Study Design	Questionnaires and Research Methods	Types of Treatment	Main Outcomes	Statistical Significance of Results
(1) Studies looking at music preferences of people with PDs or personality traits associated with PDs										
(1.1.) Studies looking at music preferences of people with PDs										
Gebhardt et al. (2014) [35]	Germany	Psychiatric patients (*n* = 180): Females (*n* = 103)*;* Males (*n* = 77). Healthy controls (*n* = 430).	610	18–82	34.6	Cross-sectional. Participants completed questionnaires to determine use of music in everyday life. Results from questionnaires analysed against reference sample.	GAF, IAAM	N/A	Patients with PDs used music mainly for cognitive problem solving and the reduction of negative activation.	T^2^-Hotelling (within analysis): F6 (*p* < 0.001). Reduction of negative activation: MANOVA ONEWAY (*p* < 0.001); cognitive problem solving MANOVA ONEWAY (*p* < 0.001).
(1.2.) Studies looking at music preferences of people with personality traits associated with PDs										
Garralda et al. (1990) [36]	UK	Children attending child psychiatric unit with emotional or conduct disorder: Females (*n* = 8); Males (*n* = 7). Clinical conduct disorder (*n* = 9); Emotional disorder (*n* = 6). High neurotic tendencies Group (*n* = 6); high antisocial tendencies Group (*n* = 4).	15	N/A	N/A	Cross-sectional. Participants’ skin conductance and heart rate changes were measured while given mental imagery tasks and while listening to music. Results were compared by diagnosis and symptom type.	RBTQ	N/A	Listening to either soft or rock music had no statistically significant difference between the groups’ cardiovascular and skin conductance reactivity.	N/A
Bowes et al. (2018) [37]	United States	North American community members: Females (46%)*;* Males (54%).	429	N/A	36.53	Cross-sectional. Participants completed surveys to determine personality traits and entertainment preferences.	PRI-R, LSRP, NPI, Mach-IV, HEXACO PI-R	N/A	Openness to experience showed a moderate association with a preference for blues and jazz music and a weak association with rock and alternative music.	Association between openness and blues and jazz (*p* < 0.001); between openness and rock and alternative (*p* < 0.001).
Gallagher et al. (2003) [38]	United States	Undergraduate psychology students: Females (67%); Males (33%). Obsessive Compulsive (OC) group (*n* = 60); Normal Control (NC) group (*n* = 60); Avoidant personality (AV) group (*n* = 40).	160	N/A	18.9	Cross-sectional. Participants sorted into OC, AV, and NC groups. Groups given cognitive ability test and factors associated with information seeking/avoidance were measured and analysed.	MR Test, OCPD DS, PDQ-4, SNAP	N/A	Participants in OC group spent less time listening to music prior to taking a stressful test compared to AV and NC groups.	Results of Fisher’s LSD comparing OC, AV, and NC groups’ time spent listening to music (*p* < 0.01)
Gerra et al. (1998) [39]	Italy	Psychosomatically healthy high school students: Females (*n* = 8); Males (*n* = 8).	16	N/A	Median: 18.6	Cross-sectional. Participants exposed to techno or classical music and changes in emotional state were measured in basal conditions and after the music exposure.	BDHI, CS, NMAC, TPQ, VZ	N/A	Novelty-seeking personality trait moderately negatively correlated with the von Zerssen score (indicating a reduced negative effect) when listening to techno music.	Association between novelty-seeking and von Zerssen Δ scores (*p* < 0.05)
Merz et al. (2021) [24]	United States	US residents recruited through online crowdsourcing platform: Females (50.3%); Males (49%).	400	18–77	34.14	Cross-sectional. Participants given questionnaires to assess music preference and aggression.	AGQ; DSM-V CCSM, RPQ, STOMP-R, NOBAGS	N/A	Preference for intense music genres (alternative, rock, punk, and heavy metal) was a nonsignificant predictor of aggression.	N/A
Sachs et al. (2021) [40]	United States	Recruited through Amazon’s Mechanical Turk (*n* = 218): undergraduates at US University (*n* = 213). Females (*n* = 431).	431	N/A	27.05	Cross-sectional. Participants given access to survey with questions presented randomly.	GEMS, IRI, RRQ, TAS, 10-IPI	N/A	Rumination was associated with liking sad music but not in positive situations (e.g., at a celebratory event). Being conscientious, emotional stability, and empathic concern were all significantly negatively correlated with using sad music in positive situations while openness to experience positively correlated with this.	Positive association between rumination and sublime feelings (*p =* 0.04); between openness to experience and “positive other” (*p* = 0.04). Negative associations between “positive other” and conscientiousness (*p* = 0.04); emotional stability (*p* = 0.03); and rumination (*p* < 0.001).
Schwartz and Fouts (2003) [41]	Canada	Junior and senior high school students: Females (*n* = 92); Males (*n* = 72).	164	12–19	16	Cross-sectional. Participants given survey to complete determining preferences for musical qualities as one of three types (heavy, light, or eclectic); music preferences were compared to each other and against measures of personality.	MAPI	N/A	Those preferring heavy music qualities were significantly more assertive in their relationships and significantly less concerned about the feelings and reactions of others. They were significantly more moody, pessimistic, sensitive, discontented, impulsive, and disrespectful to others and society.	Association between heavy music qualities and assertiveness (*p* < 0.001), social tolerance (*p* < 0.01), sensitivity (*p* < 0.01), and impulse control (*p* < 0.001). (All when compared to association between preference for light and eclectic music qualities.)
Sivathasan et al. (2021) [42]	Canada	Participants recruited through university and online: Females (*n* = 74); Males (*n* = 36).	110	18–35	21.25	Cross-sectional. Participants given surveys investigating music preferences, personality traits, and traits related to autism.	AQ, Gold-MSI-Emotion, NEO-FFI-3 (Short), SRS-SCI, SRS-2,	N/A	People with fewer autistic traits and higher levels of extraversion reported greater emotional responsiveness to music.	Association between SRS-2 and extraversion (*p* < 0.001); between NEO-E and Gold-MSI-Emotion (*p* < 0.01)
(2) Studies looking at MT for people with PDs or personality traits associated with PDs										
(2.1.) Studies looking at MT for people with PDs										
Chwalek and McKinney (2015) [43]	United States	Music therapists working in mental healthcare settings: Females (*n* = 41); Males (*n* = 6). DBT music therapists (*n* = 18); non-DBT therapists (*n* = 29).	47 + 2 music therapists recruited for qualitative interviews	23–64	39.3	Two-phase mixed methods design to evaluate the use of DBT in MT. The first phase was a quantitative online survey; the second phase was a qualitative interview.	Assessment of DBT practices	N/A	38.3% of music therapists used components of DBT in their MT practice. Most DBT music therapists used DBT with individuals with BPD, mostly to address patients’ mindfulness, emotion regulation, and distress tolerance. Few music therapists used DBT to address interpersonal effectiveness.	N/A
Foubert et al. (2017) [44]	Belgium	Psychiatric hospital patients with BPD (*n* = 16): Females (*n* = 12); Males (*n* = 3); Transgender (*n* = 1). Healthy controls from the community: (*n* = 12).	28	21–51	31	Case-control. Participants in BPD and control groups undertook MT, and their improvisations were recorded. Data analysed musically and statistically and summarised by participants’ playing style.	APD-IV, DID, ECR-R, Gold-MSI, SCID II; assessed music training/understanding.	Improvisation-based MT session using ABA structures.	In the freer improvisational section, participants with BPD did not show musical synchronicity (playing in time) compared to healthy controls.	Differences in temporal lag in section B1 between BPD participants and controls (*p* = 0.029).
Foubert et al. (2020) [45]	Belgium	Psychiatric hospital patients with BPD: Females (*n* = 15); Males (*n* = 5).	20	21–51	33	Qualitative. Participants given MT session. The principal researcher observed the improvisations and coded the data into six themes of distorted music playing.	SCID-II	Improvisation-based MT session using ABA structure.	Patients with BPD showed distorted patterns of music improvisation.	N/A
Gebhardt et al. (2018) [26]	Germany	Inpatients in psychiatric wards (PDs = 3%): Females (*n* = 85). MT Group (*n* = 82); non-MT Group (*n* = 55).	137	18–66	40.5	Cross-sectional. Participants split into MT and non-MT groups. Participants’ personality traits and use of music were measured, analysed, and compared across groups.	IAAM, SKI	Group MT: 60–85 min/once a week. (Number of sessions for participants ranged from 1–9.)	In the MT Group, insecurity predicted the use of music for cognitive problem solving and fun.	Association between reduced ego-strength and cognitive problem solving (*p* = 0.015) and fun-seeking through music (*p* = 0.025).
Hannibal et al. (2012) [46]	Denmark	Psychiatric patients with either schizophrenia (*n* = 10) or PD diagnosis *(n =* 17): Females (*n* = 15), Males (*n* = 12).	27	19–59	30	Naturalistic follow-up study measuring patients’ adherence to MT. ‘Adherence’ was defined as doing MT for longer than agreed upon.	N/A	MT: time and length of sessions varied between Participants.	Patients with PDs mostly adhered to MT (87%). None of the variables significantly predicted rates of adherence.	N/A
Kenner et al. (2020) [18]	Australia	Female outpatients of private psychiatric hospital with diagnosed BPD or traits of BPD and who had previously received DBT (*n* = 7).	7	25–60	N/A	Emergent methods qualitative study. MT sessions were filmed and then analysed. Music competency was established by authors.	N/A	MT: 75 min/once a week for 8 weeks.	Participants’ changing attitudes towards MT included increased confidence in sessions and more rhythmic synchronicity in group improvisation. Perceiving oneself as competent at group improvisation appeared useful for the broader goal of relational efficacy.	N/A
Plitt (2014) [33]	Germany	Female inpatients at psychosomatic clinic for women with BPD *(n =* 10).	10	N/A	28.5	Qualitative. MT sessions featuring improvisations and patient/therapist conversations. Improvisations and conversations were recorded and then analysed.	N/A	MT session featuring improvisations (average length 5:01 min) and conversations (average length 30:02 min).	Subjective conclusion that intersubjectivity (awareness of others and awareness of a shared experience) in MT is especially important for people with BPD.	N/A
Pool and Odell-Miller (2011) [29]	UK	Music therapists (*n* = 3); male patient with PDs included in case study (*n* = 1).	3 music therapists + 1 patient	N/A	N/A	Qualitative mixed-methods design. Single patient case study then thematic analysis of interviews with music therapists.	N/A	For case study– MT: 50 min/once a week for 10 weeks.	Subjective conclusion that aggression and creativity share important similarities in areas of control, affect, and emotion. MT can provide a context for safe exploration of aggression and other complex emotions.	N/A
Strehlow and Lindner (2016) [17]	Germany	Women who had been hospitalised by court order because of suicide attempts or suicidal tendencies: (*n* = 20)	20	19–45	(*n* = 12) < age 25	Systematic qualitative. MT sessions recorded, and scenes analysed and compared to each other. Four categories established to provide a framework to analyse scenes.	N/A	MT: 30 min/twice a week (sessions across participants ranged from 12–150).	10 themes identified as characteristic of typical BPD interactions in MT.	N/A
(2.2.) Studies looking at MT for people with personality traits associated with PDs										
Hunter and Love (1996) [47]	United States	Male inpatients at maximum security state psychiatric hospital: mentally disordered parolees (28%); mentally ill inmates (40%); patients incompetent to stand trial (12%); not guilty by reason of insanity (14%); other types of forensic and civil commitments (6%).	c. 1000	N/A	N/A	Case series. Changes implemented in hospital based on TQM methods and checked a year later if these methods had reduced mealtime violence.	TQM Methods	N/A.	Recommendations to improve mealtime safety, including having music therapists select and play music, resulted in a decrease in violent episodes at mealtimes.	Reduction of violent incidents per day a year after implementing changes (*p* < 0.001) (changes included music being played but this element could not be extracted from the other changes made).
Montello and Coons (1998) [48]	United States	Middle school students diagnosed with emotional and/or learning disturbances: Females (*n* = 2); Males (*n* = 14). Group A: received active then passive MT (*n* = 6); Group B: received passive then active MT (*n* = 4); Group C: received active MT throughout (*n* = 6).	16	11–14	11.94	Quasi-experimental. Participants assigned to either active or passive MT programmes embedded in school curriculum for two x 12 weekly sessions.	TRF	Active and passive MT: 45 min/once a week for 12 weeks (x two).	Active MT reduced Group B’s hostility scores significantly (the reverse was true in Group A, but Group A started with significantly lower hostility scores than Group B, suggesting that the groups were not matched well).	Group B (Active MT) reduction in hostility (*p* < 0.01). Group A (Active MT) increase in hostility (*p* < 0.05).
Ziv et al. (2008) [49]	Israel	Participants qualifying for diagnosis of insomnia: Females (*n* = 11); Males (*n* = 4).	15	67–93	80.63	Experimental. Participants randomly divided into two groups: first group received CD with progressive muscular relaxation (PMR); second group received CD with music relaxation method. Groups later switched interventions.	NEO PI-R LSQ, SAQ, SDQ, SSQ	PMR; music relaxation method.	The lower the agreeableness score, the greater the improvement in number of hours of sleep per night with music relaxation method. The higher the extraversion score, the greater sleep efficiency with music relaxation method.	Association between sleep length and agreeableness (*p* = 0.025); between extraversion and sleep efficiency (*p* = 0.042) (both for music relaxation method).
(3) Studies looking at music performance and personality traits associated with PDs										
Cho (2021) [50]	United States	Undergraduate music performance majors: Females (53%); Males (45%); Other (2%).	165	>70% aged 21–23	N/A	Cross-sectional. Participants completed questionnaires evaluating small ensemble attitudes/experience, personality, and empathy.	EQ, SECQ, TIPI, assessment of small ensemble experience and attitudes.	N/A	Non-classical musicians showed significantly higher empathy scores than classical musicians. Those who regularly perform in small group ensembles showed significantly higher empathy scores.	Association between non-classical musicians and EQ scores (*p* = 0.009); association between regular small ensemble playing and higher EQ scores (*p =* 0.007).
(4) Studies looking at music imagery and personality traits associated with PDs										
Mullensiefen et al. (2014) [51]	UK	Participants from UK, USA, Australia, and Europe: Females (58.1%); Males (41.4%); Gender undisclosed (0.5%). Participants used in first stage (*n* = 512); participants used in second stage (*n* = 1024).	1536	12–75	34.2	Cross-sectional. First stage: structure of musical behaviour investigated using exploratory factor analysis; second stage: data used to determine relationships between musical behaviour and subclinical OC with INMI.	OCI-R; novel questionnaire for INMI	N/A	High OC traits were positively related to INMI frequency and disturbance.	Associations between OCD-Obsessing and Frequency (*r* = 0.18) and Disturbance (*r* = 0.29); OCD-Washing and Frequency (*r* = 0.13) and Disturbance (*r* = 0.16); OCD-Neutralising and Frequency (*r* = 0.17) and Disturbance (*r* = 0.14); OCD-Ordering and Frequency (*r* = 0.15) and Disturbance (*r* = 0.15); OCD-Hoarding and Frequency (*r* = 0.15) and Disturbance (*r* = 0.19). (Correlations *r* > 0.12 are significant at 5% level.)
Negishi and Sekiguchi (2020) [52]	Japan	University students: Females (*n* = 58); Males (*n* = 43).	101	18–24	20.98	Cross-sectional. Students completed questionnaires and received phone notifications six times a day for seven days asking them to record their INMI.	Gold-MSI, J-BFI, OCTQ	N/A	There was a positive association between intrusive thoughts and the occurrence of INMI.	Association between intrusive thoughts on the occurrence of INMI: Wald z (*p* < 0.001).

Abbreviations: AGQ: Aggression Questionnaire; APD-IV: Assessment of DSM-IV Personality Disorders; AQ: Autism Quotient; AV: Avoidant Personality; BDHI: Buss-Durkee Hostility Inventory; CS: Cloninger Scale; DID: Diagnostic Inventory for Depression; DSM-V CCSM: DSM-V Level 1 Cross-Cutting Symptom Measure; ECR-R: Experiences in Close Relationships-Revised; EQ: Empathy Quotient; GAF: Global Assessment of Functioning Scale; GEMS: Geneva Emotional Music Scale; GH: Growth Hormone; Gold-MSI: Goldsmiths Musical Sophistication Index; Gold-MSI-Emotion: Goldsmiths Musical Sophistication Index Emotion; HA: Harm Avoidance; HEXACO PI-R: HEXACO Personality Inventory-Revised; IAAM: Inventory for the Measurement of Activation and Arousal Modulation; INMI: Involuntary Musical Images; IRI: Interpersonal Reactivity Index; J-BFI: Japanese Big Five Scale; LSRP: Levenson Self-Report Psychopathy Scale; LSQ: Long Sleep Questionnaire; Mach-IV: the Machiavellianism Scale-IV; MAPI: Millon Adolescent Personality Inventory; MR Test: Mental Rotations test; NOBAGS: Normative Beliefs About Aggression Scale; NC: Normal Control; NE: Plasma Norepinephrine; NEO-FFI-3 (Short): the NEO-Five Factor Inventory-3, Short Form; NEO PI-R: NEO Personality Inventory-Revised; NMAC: Nowlis Mood-Adjective Checklist; NPI: Narcissistic Personality Inventory; OC: Obsessive Compulsive; OCPD DS: Obsessive Compulsive Personality Disorder Dimensional Scale; OCI-R: the Obsessive Compulsive Inventory-Revised; OCTQ: Obsessive-Compulsive Tendencies Questionnaire; PD: Personality Disorder; PDQ-4: Personality Diagnostic Questionnaire-4; PRI-R: Psychopathic Personality Inventory-Revised; RBTQ: Rutter B Teacher Questionnaire; RPQ: Reactive-Proactive Aggression Questionnaire; RRQ: Rumination-Reflection Questionnaire; SAQ: Short Anxiety Questionnaire; SCID II: Clinical Interview for DSM-IV Axis II Disorders; SDQ: Short Depression Questionnaire; SECQ: Social-Emotional Competence Questionnaire; SKI: Self-Concept Inventory (“SelbstkonzeptInventar”); SNAP: Schedule of Nonadaptive and Adaptive Personality; SRS-2: Social Responsiveness Scale; SRS-SCI: Social Responsiveness Scale, Social Communication Index; SSQ: Short Sleep Questionnaire; STOMP-R: Short Test of Musical Preference-Revised; TAS: Tellegen Absorption Scale; TIPI: Ten-Item Personality Inventory; TPQ: Three-Dimensional Personality Questionnaire; TRF: Teacher’s Report Form; TQM: Total Quality Management; VZ Test: von Zerssen Test; β-EP: β-endorphin; 10-IPI: 10-Item Personality Index.The included studies could be categorised into four broad themes of those looking at music preference; MT; music performance; and music imagery.

**Table 2 ijerph-19-15434-t002:** Summary of main relevant results.

Category of Study	Main Relevant Conclusion(s)
Music preferences in people with PDs	Participants with PDs used music mainly for cognitive problem solving and the reduction of negative activation.
Music preference in people with traits related to PDs	Openness to experience was associated with a preference for blues and jazz music.
	Using sad music for positive reasons involving others, such as celebratory events, was positively associated with openness to experience but negatively associated with conscientiousness, empathy, and emotional stability.
	Extraversion and lower levels of traits related to autism were associated with greater emotional responsiveness to music
	In one study, a preference for heavier musical qualities was associated with increased moodiness, lower self-esteem, and increased disrespect to others.
	Novelty seekers responded less negatively to techno music compared to non-novelty seekers.
MT for people with PDs	People with BPD struggled to follow the metronomic pulse (i.e., to play in time) during music improvisation.
MT for people with traits related to PDs	Specifically chosen music may have helped to reduce mealtime aggression, violence, and hostility in a psychiatric prison population.
	Music for bedtime relaxation improved sleep length for people with lower levels of agreeableness and improved sleep efficiency for people with higher levels of extraversion.
	People with higher levels of insecurity used music to aid cognitive problem solving and for fun.
Music performance in people with traits related to PD	Non-classical music performers and those with regular small ensemble experience demonstrated higher levels of empathy than those who trained in classical music or who did not play in small ensembles regularly
Musical imagery in people with traits related to PDs	High levels of obsessive-compulsiveness were related to the frequency and occurrence of INMI.

For details see: [26,35,37,39,40,41,42,44,47,49,50,51,52].

## Data Availability

The detailed database of included studies can be requested from the corresponding author.

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
