# Peer review of "A Systematic Review of Scientific Studies on the Effects of Music in People with Personality Disorders"

_ijerph, 2022, doi:10.3390/ijerph192315434_

Round 1

Reviewer 1 Report

This manuscript is well-written on the aspect of PD and music. I am sure that It will gather a sizable amount of attention from the readers.

  I have several minor concerns that the author should address. 1. Regarding the independence of the review. There is a description "The authors independently reviewed the titles and abstracts to check whether they met the eligibility criteria", but It is not clearly stated whether two (or more) authors independently evaluated each paper. I recommend that the authors state it.   2. I think the authors should state whether they did hand-searching or not, although they did not.   3. I would like the authors to state whether they found any papers in other languages rather than English and Germany.    4. In table 2, the authors should add the first row which indicates the content of the items (i.e. country, participants' information, number, age range, mean age, etc) to make it understandable more.    5. The structure of the results should be slightly corrected. The authors divided the first section "2.4.1. Studies Looking at Music Preference in People with PDs or Personality Traits Associated with PDs" into "1. a) Studies Looking at People with PDs" and "1. b) Studies Looking at Personality Traits Associated with PDs", but they did not divide the section "2.4.2. Studies Looking at MT with People with PDs or Personality Traits Associated with PDs". They just referred to the sub-sections "Studies Looking at People with PDs" and "Studies Looking at People with Traits Relating to PDs" without 2.a) and 2. b). They should make 2.4.1 and 2.4.2 in the same way described.

Author Response

Response to Reviewer 1

This manuscript is well-written on the aspect of PD and music. I am sure that It will gather a sizable amount of attention from the readers. 

I have several minor concerns that the author should address.

  1. Regarding the independence of the review. There is a description "The authors independently reviewed the titles and abstracts to check whether they met the eligibility criteria", but It is not clearly stated whether two (or more) authors independently evaluated each paper. I recommend that the authors state it.  
  • We thank the first reviewer for this point and have clarified this information in the text.
  1. I think the authors should state whether they did hand-searching or not, although they did not.  
  • We had written in the text “the first author conducted a supplementary backward reference searching but, due to the limited literature in this area, this yielded no additional studies”. Given the first reviewer’s comment, we have added the additional line “no other hand searching was undertaken”.
  1. I would like the authors to state whether they found any papers in other languages rather than English and Germany.   
  • We thank the first reviewer for this comment and have added this information to the text: “the other reviewed studies were all in English.”
  1. In table 2, the authors should add the first row which indicates the content of the items (i.e. country, participants' information, number, age range, mean age, etc) to make it understandable more.   
  • We agree with the first reviewer that this information should be included in the results table and have added this accordingly. In addition, we have included the first author and the year of publication into the table. In accordance with the instructions to number figures and tables, we have corrected the number of the table to “Table 1”.
  1. The structure of the results should be slightly corrected. The authors divided the first section "2.4.1. Studies Looking at Music Preference in People with PDs or Personality Traits Associated with PDs" into "1. a) Studies Looking at People with PDs" and "1. b) Studies Looking at Personality Traits Associated with PDs", but they did not divide the section "2.4.2. Studies Looking at MT with People with PDs or Personality Traits Associated with PDs". They just referred to the sub-sections "Studies Looking at People with PDs" and "Studies Looking at People with Traits Relating to PDs" without 2.a) and 2. b). They should make 2.4.1 and 2.4.2 in the same way described.
  • We thank the first reviewer for noticing this inconsistency. We have amended the formatting to make the Results section correct and consistent. Please note that the numbers in the results section are now consistently start with 3.

Reviewer 2 Report

This is an interesting review about personality disorders (PD) and its relationship to music, including the references to genre, music interaction and the potential suitability of music therapy to approach PDs. There are some issues that this paper would profit in a revision.

The authors hold that they had the “intention of making the review as broad as possible”, thus sticking to a definition of PD that is incredibly unspecific. They discuss two approaches of how PDs can be classified ad at the end, it still remains unclear how exactly they define PD. One gets the impression that the conceptualization is deliberately vague so that the authors might eventually have the luxury of cherry-picking the data. Later on, they specify that “Given the limited literature published on music and PDs, the search terms were deliberately vague to capture as many studies as possible.” However, a much more concise working definition of PD would be appreciated.

Maybe this is only in the draft for peer-review but the almost empty page on p.4 should not be left “almost blank”. Chapter 2.4. says that a self-developed theoretical model (how was it developed?) “informed the development of the research question”. First, the research question should be a part of the introduction. Second, the authors failed to include a stringent research question. Instead, they simply introduced vague goals for the paper. This is not very satisfactory. Chapter 2.5. is not necessary (or it could be a side note).

Figure 1 is in very poor quality. It should be improved.

Table 2 contains valuable information. The problem is that it is rather long for being included in the main text. Authors should consider perhaps summarizing the core information in text and moving the table to the supplementary material. However, this is not so much a revision request as it is a friendly hint to the authors.

After table 2, the page formatting is wrong (i.e. the page starts counting at 1).

Table 3 is very helpful as it summarizes the main results. Many of these statements raise the suspicion that there might be a lot of confounding variables at play. For example, “Openness to experience was associated with a preference for blues and jazz music.” Maybe this was mediated by the participants’ age? I realize that – since this was a review paper – the authors might not have enough information to make a confident statement in each case if there might be a confoundedness present. However, it would be good if they could express their ideas about how they deal with the potential danger of confounded variables in the data of original articles.

The authors elaborate on the limitations of their study, which is good. They make a brief conclusion. However, could they – as researchers having now reviewed the literature – also formulate some practical advice for practitioners and clinicians?

I believe this paper is worth publishing in this journal but requires some minor revisions by addressing the above questions.

Author Response

Response to Reviewer 2

This is an interesting review about personality disorders (PD) and its relationship to music, including the references to genre, music interaction and the potential suitability of music therapy to approach PDs. There are some issues that this paper would profit in a revision.

The authors hold that they had the “intention of making the review as broad as possible”, thus sticking to a definition of PD that is incredibly unspecific. They discuss two approaches of how PDs can be classified ad at the end, it still remains unclear how exactly they define PD. One gets the impression that the conceptualization is deliberately vague so that the authors might eventually have the luxury of cherry-picking the data. Later on, they specify that “Given the limited literature published on music and PDs, the search terms were deliberately vague to capture as many studies as possible.” However, a much more concise working definition of PD would be appreciated.

  • We agree with the second reviewer that a clearer working definition of PD would assist the reader in understanding our review. We sought to undertake a broad review in an area that is currently under-researched but recognise that this needs to be balanced with providing the reader with a clear understanding of what it is we are reviewing. We have therefore clarified in the text: “As the definition, the diagnosis and the categorisation of PDs have changed over time, we included all articles into this review, where PDs were defined according to DSM-III-R, DSM-IV or DSM-5, ICD-10 or ICD-11, and all articles where the inclusion criteria into studies corresponded to the diagnostic criteria to one of the above-mentioned diagnostic manuals.”

Maybe this is only in the draft for peer-review but the almost empty page on p.4 should not be left “almost blank”.

  • We thank the second reviewer for noticing this formatting mistake and have amended it accordingly.

Chapter 2.4. says that a self-developed theoretical model (how was it developed?) “informed the development of the research question”. First, the research question should be a part of the introduction. Second, the authors failed to include a stringent research question. Instead, they simply introduced vague goals for the paper. This is not very satisfactory.

  • We thank the second reviewer for this comment. We have deleted the sentence on the theoretical model and have addressed the difficulties to formulate a research question in the discussion by adding “Additionally, because the purpose of this study was to provide an initial overview of the literature on PDs and music, we were not able to formulate a specific research question.”

Chapter 2.5. is not necessary (or it could be a side note).

  • We thank the second author for their comment and have deleted this section accordingly.

Figure 1 is in very poor quality. It should be improved.

  • We thank the second reviewer for their comment in relation to Figure 1. We have amended Figure 1 using the template provided by the PRISMA website.

Table 2 contains valuable information. The problem is that it is rather long for being included in the main text. Authors should consider perhaps summarizing the core information in text and moving the table to the supplementary material. However, this is not so much a revision request as it is a friendly hint to the authors.

  • We thank the second reviewer for their comment. We have discussed this comment and we agree that the table contains a lot of information, but we would like to keep it in the body of the text as we feel that it is key to the systematic review. We checked the recently published reviews in the IJERPH and found that most of them included the table in the main document. We have also added the authors and the publication year to the table, in keeping with other studies in the IJERPH.

After table 2, the page formatting is wrong (i.e. the page starts counting at 1).

  • We thank the second reviewer for noticing this oversight and have amended the document accordingly.

Table 3 is very helpful as it summarizes the main results. Many of these statements raise the suspicion that there might be a lot of confounding variables at play. For example, “Openness to experience was associated with a preference for blues and jazz music.” Maybe this was mediated by the participants’ age? I realize that – since this was a review paper – the authors might not have enough information to make a confident statement in each case if there might be confoundedness present. However, it would be good if they could express their ideas about how they deal with the potential danger of confounding variables in the data of original articles.

  • We thank the second reviewer for this important and helpful point. We have considered the impact of confounding variables in this review and have added a paragraph about this in the discussion section.

The authors elaborate on the limitations of their study, which is good. They make a brief conclusion. However, could they – as researchers having now reviewed the literature – also formulate some practical advice for practitioners and clinicians?

  • We thank the second reviewer for the request for recommendations or clinicians and researchers. We added two paragraphs into the discussion:

“There are several recommendations for clinicians and researchers to build and advance the research related to the use of music and music therapy with individuals with PDs. Given the unique nature of music preference, it may be helpful to conduct clinical case study research examining individuals with PDs preferred music and the functions music serves in their day to day lives [57-59]. Comparing case studies or a series of case studies can help inform best practices and foster future research recommendations. Further, conducting phenomenological studies to understand individuals with PDs experiences in MT or with different types of music interventions. This can help clinicians and researchers understand if music is being used in healthy and/or unhealthy ways [60]. Further understanding client’s experiences and uses of music and music therapy can help to inform and support in determining aims for outcome-based research.“

“From a clinical perspective, this review article identified several ways to use music in order to support people with PDs (see Table 2): As selected music was reported to help to reduce violence and hostility, patients may develop playlists with the support of their therapists to manage aggression and violent impulses in foreseeably difficult sit-uations; appropriate music for bedtime relaxation can be recommended to improve sleep length and quality; and people who experience insecurities may be encouraged to use music to aid cognitive problem solving and to improve their mood.”

I believe this paper is worth publishing in this journal but requires some minor revisions by addressing the above questions.

Reviewer 3 Report

Dear authors,

your reasearch is excellent, it is very profound, up-to-date and for me, very interesting. It can be drawn upon in the future. I find music very important part of humans´ life, not only when someone suffers from PD. In those cases, music can be very benfitial. For the future, I would recommend to consider to include research questions in your papers and to draw conclusions based on these questions - it would give your work more compact design. 

I wish you lots of ispiration and energy for future work.

Author Response

Response to Reviewer 3

  • We thank the third reviewer for their generous and supportive comments. We agree that a future review would benefit from more specific research questions, which will hopefully be possible when there is a greater wealth of literature in this area. We will take this into consideration for future research. We have also added the following to our limitations: “Additionally, because the purpose of this study was to provide an initial overview of the literature on PDs and music, we were not able to formulate a specific research question.”